# PTB Regulates the Metabolic Pathways and Cell Function of Keloid Fibroblasts through Alternative Splicing of PKM

**DOI:** 10.3390/ijms24065162

**Published:** 2023-03-08

**Authors:** Rong Huang, Rong Han, Yucheng Yan, Jifan Yang, Guoxuan Dong, Miao Wang, Zhiguo Su, Hu Jiao, Jincai Fan

**Affiliations:** The Plastic Surgery Hospital, Chinese Academy of Medical Sciences and Peking Union Medical College, No. 33 Ba-Da-Chu Road, Shijingshan District, Beijing 100144, China

**Keywords:** keloid, fibroblast, cell metabolism, Warburg effect, PTB, PKM2

## Abstract

Keloids, benign fibroproliferative cutaneous lesions, are characterized by abnormal growth and reprogramming of the metabolism of keloid fibroblasts (KFb). However, the underlying mechanisms of this kind of metabolic abnormality have not been identified. Our study aimed to investigate the molecules involved in aerobic glycolysis and its exact regulatory mechanisms in KFb. We discovered that polypyrimidine tract binding (PTB) was significantly upregulated in keloid tissues. siRNA silencing of PTB decreased the mRNA levels and protein expression levels of key glycolytic enzymes and corrected the dysregulation of glucose uptake and lactate production. In addition, mechanistic studies demonstrated that PTB promoted a change from pyruvate kinase muscle 1 (PKM1) to PKM2, and silencing PKM2 substantially reduced the PTB-induced increase in the flow of glycolysis. Moreover, PTB and PKM2 could also regulate the key enzymes in the tricarboxylic acid (TCA) cycle. Assays of cell function demonstrated that PTB promoted the proliferation and migration of KFb in vitro, and this phenomenon could be interrupted by PKM2 silencing. In conclusion, our findings indicate that PTB regulates aerobic glycolysis and the cell functions of KFb via alternative splicing of PKM.

## 1. Introduction

Keloids are unsightly rubbery lesions that appear on the skin as a result of poor wound healing [1]. The typical keloid is significantly impacted by hereditary factors, which helps to explain the ethnic disparities in susceptibility to the formation of keloids: they are most common in Africans (5–10%), less common in Asians (0.1–1%), and uncommon in Europeans/North Americans (<0.1%) [2]. As well as causing cosmetic disfigurement, keloids can be painful and pruritic, leading to intense psychological stress [3]. Studies have reported that pain affects 55.7% of keloid patients, whereas pruritic symptoms affect 72.3%. Keloids recur even after a successful treatment strategy. Surgical removal, in particular, has a high recurrence rate (45–100%) [4]. Response rates to single intralesional triamcinolone injections range from 50% to 100% clinical efficacy, with recurrence rates of 9% to 50% [5]. Other treatments, including pressure therapy, cryosurgery, and others, are not commonly used because of their limited success and unfavorable impacts [6]. The fundamental reason for the difficulty of treatment lies in the poorly understood etiology. In recent years, although scholars have revealed that inflammation, immunity, genetics, and epigenetics are all involved in the mechanisms of keloids, a thorough understanding of their mechanisms has yet to be achieved [7,8,9].

Intensive investigations in the field of metabolism have proven that aerobic glycolysis is mostly responsible for the oncogenic lesions in cancer cells, and metabolic reprogramming has progressed well beyond expectations [10]. Recent evidence has revealed that metabolic disturbances also occur in keloids. In vitro, keloid fibroblasts (KFb) displayed an antiproliferative effect and a decrease in extracellular matrix protein after treatment with the antimetabolic chemical rapamycin. Additionally, Liu [11] showed that keloids upregulate the expression of genes involved in mitochondrial oxidative phosphorylation. Furthermore, Li [12] discovered that KFb underwent metabolic phenotypic reprogramming via oxidative phosphorylation and aerobic glycolysis with an augmented glycolysis capacity. Moreover, our previous study first confirmed that the Warburg effect is unique to keloids and has not been found in other types of scars [13]. The Warburg effect was proposed by Otto Warburg in the 1920s to describe how tumor cells switch their metabolism from oxidative phosphorylation to glycolysis even in the presence of ample oxygen, a process he termed “aerobic glycolysis” [14,15]. Recent studies have shown that the Warburg effect substantially impacts many characteristics of tumor cells, such as proliferation, apoptosis, autophagy, angiogenesis, tumor stem cells’ epithelial–mesenchymal transformation, and formation of the microenvironment [16,17]. However, the exact influence of the Warburg effect on keloids and the exact regulation mechanism of the Warburg effect in keloids is not yet known.

During the process of the Warburg effect, pyruvate kinase (PK) functions as a crucial molecule and has four varied isoforms, which are PKM1, PKM2, PKL, and PKR [18]. PKM2, not PKM1, has a property that catalyzes the signaling cascades initiated by the cell growth factor and controls the glycolytic metabolism [19,20]. The expression of this isoform is found in a variety of malignancies [21,22]. In tumor cells, after siRNA treatment of PKM2, the Warburg effect was inhibited, and cells executed apoptosis or autophagy [19,23]. PTB protein (polypyrimidine-tract-binding, also known as PTBP1/HNRNP I), functions as an alternative splice site repressor by binding preferentially to the polypyrimidine tract [19,24]. The role of PTB in alternative splicing sites has been previously connected with malignant transformation in cancer cells [25,26,27]. PTB dominantly selects Exon 10, resulting in the critical switch of PKM1 to PKM2 [19]. Our past study observed that PTB is also elevated in KFb, and the growth of KFb is inhibited after the knockdown of PTB [28], which reminded us of the important role of PTB in keloids. Interestingly, many studies have demonstrated that PTB can regulate the Warburg effect. The main focus of this study was to identify the potential mechanisms responsible for this hyperglycolytic phenotype of keloids and the potential role of PTB in these changes, which has not been elucidated previously.

Here, we demonstrated that metabolic disturbance of keloids is strongly associated with the upregulation of PTB. Furthermore, the PTB-regulated PKM splicing of PKM1 to PKM2 enhances the Warburg effect of KFb. On the basis of this metabolic finding, we further revealed that PTB promotes the proliferation and migration of keloids via alternative splicing of PKM. Such metabolic findings promise a powerful tool for diagnosing and treating keloids. In addition, we explored the link between PTB and the tricarboxylic acid (TCA) cycle.

## 2. Results

### 2.1. The Expression of PTB Was Abnormally Elevated in Keloid Samples from Patients

To examine the expression of PTB in keloids and the surrounding normal skin, we first performed an IHC assay on four pairs of matched keloids and corresponding normal skin tissue samples. The IHC of PTB confirmed the localization of the PTB of KFb in the superficial and deep dermis of the keloid, with positive staining also observed in the epidermis (Figure 1A). Most of the keloid tissues (75%, 3/4 cases) showed high levels of PTB expression, while 75% (3/4 cases) of the corresponding normal skin tissues exhibited low levels of PTB expression (Figure 1B). Collectively, these data indicated that PTB is frequently upregulated in keloids. Cellular immunofluorescence showed that the nucleus was substantially enriched in PTB (Figure 1C). In the four pairs of KFb and the corresponding NFb samples, the mRNA expression level of PTB was significantly upregulated in KFb compared with NFb except in Case 4, which exhibited the opposite result: higher PTB levels in NFb but lower levels in KFb (Figure 1D). The expression levels of PTB protein in the fibroblasts measured by immunoblotting were consistent with the mRNA expression data (Figure 1E). Therefore, PTB expression was upregulated both in keloid tissues and KFb, and thus PTB may serve as a marker for the diagnosis of keloids.

### 2.2. PTB Promotes Glycolysis Flux in KFb

To explore the role of PTB in aerobic glycolysis, KFb was transfected with siRNA for the purpose of silencing PTB, and NFb was transfected with a lentivirus vector to overexpress PTB. Through the use of qRT-PCR and Western blotting, it was proven that PTB knockdown and overexpression were effective (Figure 2A–D). As expected, silencing PTB decreased the lactate production (Figure 2E) and glucose uptake (Figure 2F) of KFb. PTB knockdown resulted in a 54.06 ± 0.05% decrease in lactate production and a 33.42 ± 0.01% decrease in glucose consumption in KFb. In addition, the overexpression of PTB in NFb led to increased lactate production (Figure 2G) and glucose uptake (Figure 2H). Overexpression of PTB led to a 21.93 ± 0.02% increase in lactate production and a 47.38 ± 0.06% increase in glucose consumption.

### 2.3. PTB Promotes Key Glycolysis Enzymes in KFb

We also measured the mRNA and protein levels of the critical enzymes responsible for cell metabolism, namely glucose transporter proteins (GLUT) and lactate dehydrogenase A (LDHA). It was documented that the level of GLUT1 was elevated in KFb compared with NFb under the same oxygen conditions [29]. LDHA was found to be elevated in KFb in our past study [13]. In this study, we confirmed the downregulation of GLUT1 and LDHA following PTB knockdown, while both qRT-PCR and Western blot analysis revealed that GLUT3 only changed a little following PTB knockdown (Figure 3A–E). On the contrary, both GLUT1 and LDHA were upregulated after the overexpression of PTB, whereas GLUT3 showed no significant change after the overexpression of PTB (Figure 3F–J). After knockdown of PTB in KFb, we examined the levels of hexokinase II (HKII) and 6-phosphofructo-2-kinase (PFKFB3), which are two key rate-limiting enzymes in glycolysis, and found a significant decrease in HKII and PFKFB3 after the silencing of PTB in KFb, via both qRT-PCR (Figure 3K,L) and Western blot detection (Figure 3M). On the contrary, the overexpression of PTB increased the expression of HKII and PFKFB3 both in terms of the mRNA (Figure 3N,O) and protein levels (Figure 3P).

### 2.4. PTB Regulates PKM Splicing in KFb

PTB is a critical splicing factor determining the relative expression of PKM1 and PKM2, which are vital rate-limiting enzymes of glycolysis [19]. We first determined the balance between PKM1 and PKM2 in keloid patients. When keloid tissues were compared with matched normal skin tissues, we found that the expression of PKM2 was considerably higher and the expression of PKM1 was significantly lower when using an IHC assay of PKM2 and PKM1 (Figure 4A–D). When keloid tissues were compared with matched normal skin tissues, we found half of the normal tissues (50%, 1/2 cases) expressed high PKM1 levels and the remaining samples expressed low PKM1 levels. When it came to the keloid tissues, 75% (3/4 cases) expressed low PKM1 levels. Unlike the results for PKM1, all of the keloid tissues expressed high levels of PKM2 while only half of the normal tissues expressed high levels of PKM2.)

Furthermore, RT-PCR and restriction digestion were used to analyze the mRNA levels of PKM1 and PKM2. We observed that the mRNA levels of PKM2 increased and the mRNA levels of PKM1 decreased in KFb compared with the corresponding NFb (Figure 4E) (*n* = 3). The relative expression of te PKM1 and PKM2 protein isoforms in the keloid was region-specific: KFb preferentially expressed PKM2, whereas the adjacent NFb mainly expressed PKM1, which was measured by immunoblotting using isoform-specific antibodies (Figure 4F) (*n* = 4).

In addition, we examined the expression of PKM following the silencing of PTB in KFb and demonstrated the decreased expression of PKM2 protein at 72 h post-transfection (Figure 4G). On the contrary, the overexpression of PTB in NFb led to an increase in PKM2 protein levels (Figure 4H). These observations demonstrated that PTB plays a role in inducing the transformation from PKM1 to PKM2. It should be noted that the protein levels of PKM1 increased only slightly as a result of PTB downregulation, while the protein levels of PKM2 remained stable.

### 2.5. PTB Regulates Aerobic Glycolysis through Alternative Splicing of PKM

We knocked down PKM2 in KFb through PKM2 siRNA (siPKM2), and the efficiency of silencing was detected by qRT-PCR (Figure 5A) and Western blotting (Figure 5B). The data showed that lactate production (Figure 5C) and glucose uptake (Figure 5D) also decreased in KFb transfected with siPKM2. PKM2 knockdown led to a 41.54 ± 0.03% decrease in lactate production and a 56.44 ± 0.03% decrease in glucose uptake. To examine the specific mechanism of PTB and PKM2 in regulating the Warburg effect, we knocked down PKM2 with siPKM2 while overexpressing PTB in NFb simultaneously, and then measured the glycolysis influx. The effects of overexpressing PTB and PKM2 silencing were confirmed by Western blotting (Figure 5E) and qRT-PCR (Figure 5F,G). We observed that the knockdown of PKM2 greatly reduced the tendency of PTB to enhance lactate production (Figure 5H) and glucose uptake (Figure 5I). These findings demonstrate that PTB enhances the aerobic glycolysis of KFb by maintaining a high level of PKM2.

### 2.6. PTB Knockdown Downregulated the Key Enzymes of the TCA Cycle

The TCA cycle and glycolysis are connected by the rate-limiting enzyme PKM2. The relationships between PTB or PKM2 and the key enzymes in the TCA cycle, however, have not received much attention. Here, using qRT-PCR, we found that the mRNA expression of oxoglutarate dehydrogenase (OGDH) (Figure 6A) decreased markedly with PTB silencing in KFb, whereas upregulation of the mRNA levels of citrate synthase (CS) (Figure 6B) was observed. As for isocitrate dehydrogenase1 (IDH1) (Figure 6C), there was no significant difference between the si-NC and si-PTB groups at the mRNA level. However, silencing PTB decreased the protein levels of these three enzymes’ (Figure 6D). Moreover, the mRNA expression levels of OGDH (Figure 6E) and CS (Figure 6F) were elevated, while the mRNA levels of IDH1 (Figure 6G) decreased with the knockdown of PKM2. Furthermore, in the PKM2-silenced KFb, the protein levels of CS and IDH1 decreased while those of OGDH increased (Figure 6H). The mRNA levels and the protein changes of each enzyme varied greatly. Complex post-transcriptional processing might contribute to this. In brief, PTB might augment the TCA cycle’s adaptation to the microenvironment of KFb.

### 2.7. PKM2 Knockdown Abolished the PTB-Induced Proliferation and Migration of Fibroblasts

We investigated the change in the proliferation of KFb under PTB regulation using the CCK-8 assay. At 48 h after transfection with siRNA or lentivirus, the fibroblasts were seeded at a density of 3000 cells per well into 96-well plates and cultivated for five days. As expected, KFb treated with si-PTB demonstrated significantly lower growth rates at 48 h, and maintained this at 120 h, compared with KFb transfected with si-NC (Figure 7A). On the contrary, NFb overexpressing PTB had upregulated cell proliferation ability (Figure 7B). In addition, KFb treated with si-PKM2 demonstrated significantly lower growth rates at 48 h, and maintained these at 120 h, compared with the KFb transfected with si-NC (Figure 7C). On the contrary, the overexpression of PKM2 in NFb promoted cell proliferation (Figure 7D). We next sought to determine the role of PKM2 in PTB’s enhancement of proliferation. The data showed that the knockdown of PKM2 lessened the effect of enhanced proliferation caused by PTB (Figure 7E).

In addition to their ability of proliferation, another growth characteristic of keloids is their strong migration ability. Thus, we identified the role of PTB in the migration ability of KFb using a scratch wound assay. Specifically, silencing PTB suppressed the migration of KFb (Figure 7F). A scratch wound assay showed that KFb transfected with si-NC migrated from the edges of the scratches into blank areas, with nearly full closure on the second day, while on Day 2, KFb transfected with si-PTB showed slower cell proliferation, reduced migration, and incomplete coverage of the unoccupied regions. However, the overexpression of PTB enhanced the cell migration of NFb (Figure 7G). Similarly, the migration ability of KFb was significantly inhibited following PKM2 silencing in KFb (Figure 7H), while the overexpression of PKM2 in NFb could significantly increase the migration rate (Figure 7I). Additionally, we discovered that the enhancement in cell migration caused by overexpression of PTB was eliminated by knocking down PKM2 (Figure 7J). Taken together, these results suggest that PTB induced increased proliferation and migration of KFb in vitro through PKM2.

## 3. Discussion

Numerous studies have documented that KFb’s metabolic profile is reprogrammed from oxidative phosphorylation through the Warburg effect. We further discovered, in a previous study, that the Warburg effect is exclusive to KFs and is not found in other types of scars. Hence, keloids have a unique energy metabolism compared with other types of scar. However, the potential mechanism of the Warburg effect is poorly understood. Here, we measured the level of PTB, a molecule associated with metabolic reprogramming, as well as the changes in glycolysis following the regulation of PTB.

Molecules associated with metabolic reprogramming have been studied in cancer cells. The expression of PTB is upregulated in many cancers, and it can regulate the splicing patterns related to the pathogenesis of numerous diseases [30]. Research has demonstrated that inhibiting PTB reduces immune surveillance without raising the risk of carcinogenesis, and weakens the protumorigenic consequences of the senescence-associated secretory phenotype [31]. This may show that the knockdown of PTB is safe and could be used to treat proliferative diseases. However, studies have also reported that the knockdown of PTB increased the ability of HT29 cancer cells to invade other tissues [32,33]. This would suggest that PTB functions differently in different tissues, and therefore, more accurate targeting for treating disease is needed. Consistent with most reported findings, we confirmed the upregulation of PTB in keloid tissues compared with that in the surrounding normal skin tissues.

Cellular metabolism is essential for fibroblasts’ phenotype and function [34]. Previous research has suggested that despite the massive demand for oxygen in keloid fibroblasts, these cells do not rely on the oxidative metabolism but mainly on glycolysis for their ATP requirements [35]. Various miRNAs targeting PTB alter the cells’ glucose metabolism and contribute to the Warburg effect [36], suggesting a mechanistic link between the expression of PTB and metabolic dysfunction. In the present study, we explored the glycolysis flux and key glycolysis enzymes in the KFb by knocking down PTB. When we silenced PTB, our data showed a decrease in glucose uptake and lactate generation in KFb. On the contrary, increased glucose uptake and lactate generation occurred when the PTB was overexpressed in NFb. Although glucose in aerobic glycolysis generates less ATP than the amount synthesized by oxidative phosphorylation, the rate of glucose metabolism through the enhanced aerobic glycolysis induced by PTB is much higher. LDHA acts as a critical enzyme to catalyze the conversion of pyruvate into lactate, and GLUT mediates the central step in limiting the utilization of intracellular glucose [37]. The investigated mRNA and protein expression levels of GLUT1 and LDHA are consistent with the change in glycolysis flux following a change in the levels of PTB. In addition, we further explored the expression of PFKFB3 and HKII and showed that the level of these two key rate-limiting enzymes of glycolysis also decrease following the knockdown of PTB. As the vital rate-limiting enzymes in glycolysis, PFKFB3 and HKII are highly associated with tumorigenesis, with preferential expression [38,39]. Apart from the regulation of pKM by PTB, few studies have focused on the regulatory effect of PTB on other key enzyme, which also merits further investigation.

It is well documented that cancer cells are prone to utilizing glycolysis as a major catabolic pathway to produce energy and using anabolic precursors to meet the requirements of their survival and rapid growth [40]. Thus, inhibiting the glycolytic process is thought of as a potential cancer treatment method [41]. The literature has reported that the upregulation of PKM2 and downregulation of PKM1 contribute to increased aerobic glycolysis, consequently increasing cell proliferation in cancer cells [18,19]. Our study demonstrated that the phenomenon of regulated alternative splicing of PKM by PTB also occurs in KFb. PTB binds to Exon 10 of PKM2 in a steady state; however, PTB also binds to Exons 9 and 10 after treatment with PTB siRNA [42]. KFb dominantly expresses PKM2 and thereby predominantly performs glycolysis, even in the presence of enough oxygen. The promotion of glycolysis induced by PKM2 in KFb results in an increased release of lactate. Furthermore, the knockdown of PKM2 greatly diminishes the increased lactate effect of glycolysis induced by the overexpression of PTB. Taken together, our study demonstrated that PTB enhances aerobic glycolysis by regulating the splicing of PKM.

It should be noted that the PKM1 protein increased only slightly with the downregulation of PTB, whereas the PKM2 protein remained almost stable. Similarly, the knockdown of hnRNP A1 and hnRNP A2 allowed for only a small increase in PKM1 [18]. Thus, we hypothesized that other elements might also result in the strict regulation of the alternative splicing of PKM. PKM genes are likely to be under the combined splicing regulation of multiple splicing regulators, as in many other examples of controlled alternative splicing [43]. In addition, interestingly, we suspected that the changes in lactate metabolism after PTB silencing are probably caused by more than just the PKM splicing phenomenon. One study reported that a complete switch from PKM2 to PKM1 could reduce the generation of lactate by up to 30% [20], and the drop we observed after the knockdown of PKM2 was close to 42%. However, our study showed that the knockdown of PTB caused a decrease of nearly 54% in lactate production, which could be a reflection of other alternative splicing process that likely happen in addition to PKM splicing. Considering that PFKFB3 also has multiple alternatively spliced isoforms [39] and is also under the regulation of PTB, we speculated that the regulation of PTB on the alternative gene splicing of PFKFB3 also occurred, which would explain the considerable overall reduction in lactate generation observed after knocking down PTB.

In order to fulfill the high demand for biosynthesis during proliferation, proliferation cells increase glycolysis, which helps to pool glycolytic intermediates and limit the amount of carbon that may pass from glucose to the TCA cycle [10]. However, some scholars doubt that a change in glycolysis alone is sufficient for cells with high metabolic requirements for proliferation [44]. Thus, we further examined the influence of the knockdown of PTB and PKM2 on the TCA cycle in keloids, comparing the critical enzymes changes after the downregulation PTB or PKM2. Interestingly, we found that the knockdown of PTB could significantly inhibit the essential enzymes, and a similar phenomenon also occurred with a knockdown of PKM2 during protein detection, whereas change in the mRNA is not always consistent with the protein expression levels. Complex post-transcriptional and translational regulation may occur and then cause the different protein and mRNA levels. It has been reported that the knockdown of PKM2 could lead to a decrease in glutamine and lactate, which are both primary carbon sources [44,45,46]. Thus, we believe that reducing the carbon sources following the knockdown of PTB or PKM2 inhibited the expression of enzymes in the TCA circle. However, one study reported that the switch from PKM2 to PKM1 led to the activation of the TCA cycle [36]. These different reports remind us to further explore the specific metabolic role of PKM2 in connecting glycolysis and TCA to help us better understand its regulation of the metabolic budgeting system.

The characteristic of aggressive growth beyond the border of keloids is a primary anxiety for patients; thus, exploring the molecular mechanisms that drive the progression of keloids was necessary. Our data revealed that the knockdown of PTB inhibited the hyperproliferative and migration phenotype of KFb. On the contrary, the overexpression of PTB enhanced the proliferation and migration capability of NFb. We further found that PKM2 silencing abolished the PTB-induced proliferation and migration in KFb. It is believed that growing cells can incorporate metabolites from glycolysis for the synthesis of macromolecules for cell growth, which might further contribute to cell proliferation [47,48]. Therefore, we are convinced that the enhanced glycolysis induced by PTB might further enhance the proliferation and migration of KFb.

## 4. Materials and Methods

### 4.1. Tissue Samples and Cell Culture

Normal skin samples were obtained from patients who underwent cosmetic surgery, while the keloid tissue samples were harvested from patients who had undergone surgery for keloid removal. There were four pairs of matched keloid and corresponding normal skin tissues among them. Pre-surgical keloid therapy was not applied to the chosen samples. In Table 1, the information of the samples is provided. Briefly, the keloid or normal skin tissue was rinsed with sterile PBS 5–10 times, then the tissue was immersed in a mixture of penicillin and streptomycin for 30 min. After that, the tissue was placed in a sterile bending plate, and the subcutaneous adipose tissue was removed with a sterile scalpel. The remaining tissue was repeatedly cut into particles of about 1 mm^3^. Finally, the tissue particles thus obtained were transferred to a 25 mL sterile culture bottle and cultured in an incubator containing 5% CO_2_ at 37 °C. When the fibroblasts migrated from the tissue and covered about 70% of the bottom of the culture bottle, 0.25% trypsin was used to digest the fibroblasts for 1 to 3 min. A total of 1 mL of Dulbecco’s Modified Eagle’s Medium (DMEM) (HyClone, Logan, UT, USA) containing 10% fetal bovine serum (FBS) (Gibco, Waltham, MA, USA) was then used to terminate digestion. The cell suspension was collected in a 15 mL centrifuge tube and centrifuged for 5 min at 1000 rpm. The supernatant was then discarded, and the cell mass was suspended and cultured in an incubator. Cells at 2–3 passages were prepared for experimental use. This study was approved by the ethics committee of The Plastic Surgery Hospital of the Chinese Academy of Medical Sciences and followed the principles stipulated in the Declaration of Helsinki. Patients signed consent forms agreeing to their data being used and analyzed.

### 4.2. Immunohistochemical (IHC) Staining and Scoring Analyses

Briefly, harvested tissue went through fixation, paraffin embedding, and cut selection. Next, paraffin selections were deparaffinized and hydrated. The slides were then treated with 0.3% H_2_O_2_ to inhibit endogenous peroxidase activity, followed by heat antigen retrieval in a citrate buffer (pH 6.0). The tissues were incubated with primary antibodies for one night at 4 °C and then the species-appropriate secondary antibodies for 1 hour at room temperature. Display staining was performed with a DAB kit (Beyotime, Shanghai, China). Two independent pathologists blinded to the samples’ information examined all the IHC results. Protein levels (score) were determined on the basis of the staining intensity and staining location using a semiquantitative scoring system [49]. Scores of <4 indicated low protein expression levels, and scores of ≥4 indicated high protein expression levels.

### 4.3. RNA Interference

We used the following short interfering RNA (siRNAs) (TsingKe, Beijing, China) to carry out RNA interference: PTB siRNA, 5′-GCAAGAAGUUCAAAGGUGAdTdT-3′; PKM2 siRNA, 5′-CCAUAAUCGUCCUCACCAATT-3′; and negative control siRNA. Briefly, we plated fibroblasts at 3 × 105 cells per well in 6-well plates. The next day, 50 pmol of siRNA with 4 µL of jetPRIME (Polyplus Transfection, Illkirch-Graffenstaden, France) plus 200 µL of the buffer was mixed and added to the fibroblasts. RNA and protein were collected at 48 h and 72 h, respectively, after transfection to perform the analyses.

### 4.4. Lentivirus Production and Overexpression of PTB and PKM2 in NFb

The full-length sequences of PTB and PKM2 were amplified by PCR and cloned into the pCDH-CMV-MCS-EF1-copGFP vector to construct the recombinant plasmids pCDH-PTB and pCDH-PKM2. The target plasmid, psPAX2, and pMD2.G plasmids were mixed and packaged into 293T cells to produce the lentivirus. Normal fibroblasts (NFb) were transfected with an overexpression lentivirus (PTB or PKM2) and an empty lentivirus using 6 µg/mL polybrene (Merck Millipore, Darmstadt, Germany).

### 4.5. RNA Isolation and qRT-PCR

Cellular RNA was isolated with TRIzol reagent (TaKaRa, Kusatsu, Japan). Next, a NanoDrop 2000 spectrophotometer device (Thermo Fisher Scientific, Waltham, MA, USA) was used for detection of the purity and concentration. RNA was transcribed into cDNA using the PrimeScript RT Reagent Kit (Takala, Japan). qRT-PCR was conducted with a standard SYBR Green PCR kit protocol (Roche, Basel, Switzerland) and performed in a LightCycler 96 Real-Time System (Roche, Switzerland). The expression levels of β-actin were used to normalize the results. The primer sequences used are displayed in Table 2.

### 4.6. Western Blotting

Fibroblasts were extracted in a RIPA buffer (Beyotime, China) containing protease inhibitor cocktails (Solarbio, Beijing, China), then the supernatant of the lysate was harvested after centrifugation and its concentration was determined with a BCA Protein Assay kit (Beyotime, China). Protein samples were resolved by 4–12% SDS-PAGE and transferred to PVDF membranes with 0.45 µm pores (Millipore, Burlington, MA, USA). After being blocked with non-fat milk powder, the membranes were incubated with primary antibodies for one night. The secondary antibodies were goat anti-mouse-IgG-HRP (1:1500, Proteintech, Rosemont, IL, USA) and goat anti-rabbit-IgG-HRP (1:5000, Abcam, Waltham, MA, USA). Proteins were detected with a chemiluminescence (ECL) kit (Millipore, Burlington, MA, USA). The Western blotting bands were visualized by the ChemiDoc MP system (Bio-Rad, Hercules, CA, USA). The endogenous reference for normalization was β-actin. Specific information about the antibodies used can be found in Table 3.

### 4.7. CCK-8 Assay

A Cell Counting Kit-8 (Beyotime, China) was used to detect the fibroblasts’ proliferation. At 48 h post-transfection with siRNA or the lentivirus, fibroblasts were seeded onto 96-well plates at a density of 3000 cells per well. An enzyme-labeled device (Thermo Fisher Scientific, Saint Louis, MO, USA) was used to measure the absorbance at 450 nm at the indicated time points.

### 4.8. Scratch Wound Assay

To investigate the migration of the fibroblasts, a scratch wound assay was used. In total, 2 × 10^5^ fibroblasts per well were planted onto 6-well plates. At 48 h after transfection with siRNA or the lentivirus, and the cells had reached nearly 100% confluence, a scratch wound was made using a 200 µL tip on the monolayer of cells. The cells were given three PBS washes before being given DMEM for culturing. Digital photographs of each wound were taken under a Nikon Eclipse E200 microscope (Nikon, Tokyo, Japan) both instantly (0 h) and 24 h following after development of the scratch. Images were captured from three arbitrary perspectives. Image J software was used to analyze cell migration.

### 4.9. Immunofluorescence

The cells were permeabilized with 0.1% Triton X-100 for 5 min at room temperature after being fixed with 4% paraformaldehyde for 20 min. The cells were treated with the corresponding primary antibodies overnight at 4 °C after being blocked with 1% BSA for 30 min. After being washed with PBS for 15 mins, goat anti-rabbit IgG secondary antibody (Abcam, USA) that was labeled with Alexa Fluor 488 was incubated with the cells. The cells’ nuclei were dyed with DAPI for 1 min before being examined under a confocal microscope (ZEISS, Oberkochen, Germany).

### 4.10. Glucose Uptake and Lactate Production

First, the fibroblasts were transfected with the lentivirus or siRNAs in a culture medium for 1 day. Next, the medium was changed to 1640 medium devoid of phenol red for 2 days.

The culture medium was collected from each sample to detect glucose uptake and lactate production using the glucose colorimetric assay kit (BioVision, Milpitas, CA, USA) and the lactate colorimetric assay kit (BioVision, USA) following the manufacturer’s instructions.

### 4.11. RT-PCR and PKM Splicing Assays

Total cell or tissue RNA was extracted using TRIzol reagent (TaKaRa, Japan). After reverse transcription of the RNA to cDNA, the PCR products of PKM were digested using PstI, then the digested mixtures were separated using 1% agarose gel.

### 4.12. Statistical Analyses

The data analyses was primarily conducted using GraphPad Prism 5.0 (GraphPad Software, San Diego, CA, USA). Student’s *t*-test was performed to identify significant differences between the two groups. Data are reported in this article as the mean ± standard deviation for at least three independent experiments, and *p* < 0.05 was considered to be significant (* *p* < 0.05; ** *p* < 0.01; *** *p* < 0.001; **** *p* < 0.0001).

## 5. Conclusions

In summary, we have provided evidence for the central role of upregulated PTB in the enhanced glycolysis of KFb in keloid patients. The downregulation of PTB levels in vitro restored the regular expression of PKM2 and other glycolysis influx genes, which reversed the hyperproliferation and migration of KFb. In addition, the knockdown of PTB and PKM2 decreased the expression of the vital rate-limiting enzymes in the TCA cycle.

## Figures and Tables

**Figure 1 ijms-24-05162-f001:**
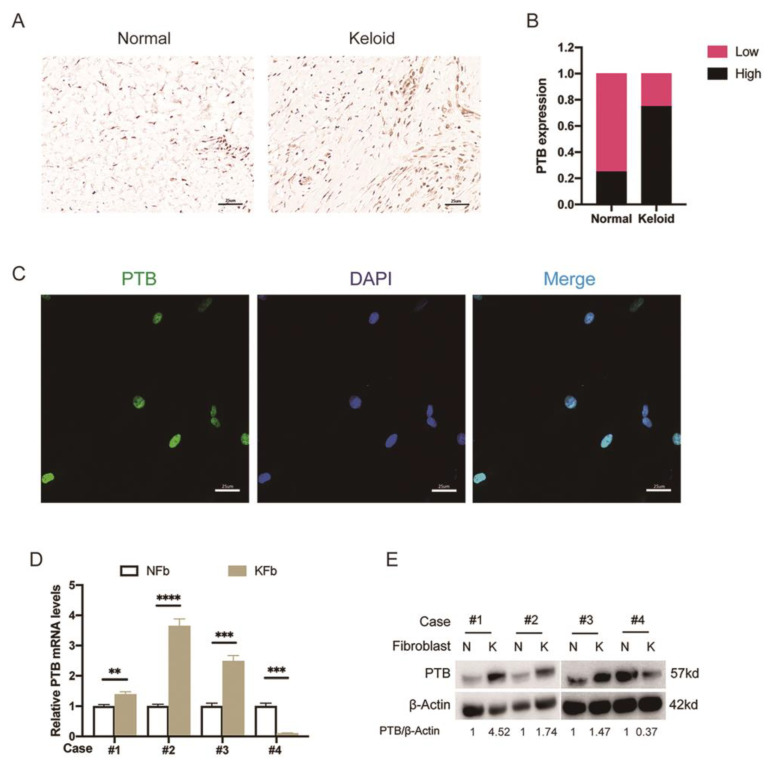
The aberrantly upregulated PTB in keloid samples. (**A**) The expression of PTB was detected in keloid tissues and the adjacent normal skin tissues by IHC. Representative images are shown at 400× magnification. (**B**) Differences in the expression scores of PTB between keloid tissues and the corresponding normal tissues are presented as a histogram (*n* = 4). (**C**) Confocal microscopy was used to observe KFb immunostained with anti-PTB antibody. (**D**) PTB mRNA levels in the KFb (*n* = 4) and the matched adjacent NFb (*n* = 4) were detected by qRT-PCR. (**E**) PTB protein levels in the KFb (*n* = 4) and matched adjacent NFb (*n* = 4) were detected by Western blotting. Data are presented as the mean ± S.D. ** *p* < 0.01, *** *p* < 0.001, **** *p* < 0.0001.

**Figure 2 ijms-24-05162-f002:**
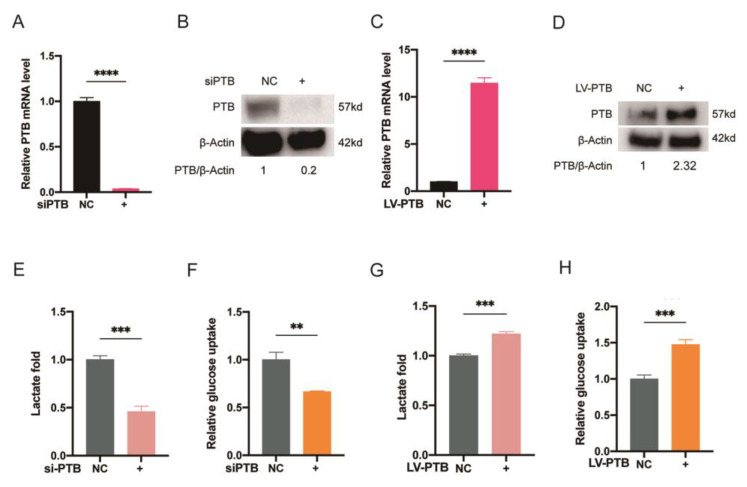
PTB promotes glycolysis flux in KFb. (**A**,**B**) The knockdown of PTB in KFb was measured by qRT-PCR and Western blotting. (**C**,**D**) Overexpression of PTB in NFb was measured by qRT-PCR and Western blotting. Lactate production (**E**) and glucose uptake (**F**) were measured in the KFb transfected with si-PTB. Lactate production (**G**) and glucose uptake (**H**) were measured in the NFb transfected with the PTB lentivirus vector. (*n* = 3). ** *p* < 0.01, *** *p* < 0.001, **** *p* < 0.0001.

**Figure 3 ijms-24-05162-f003:**
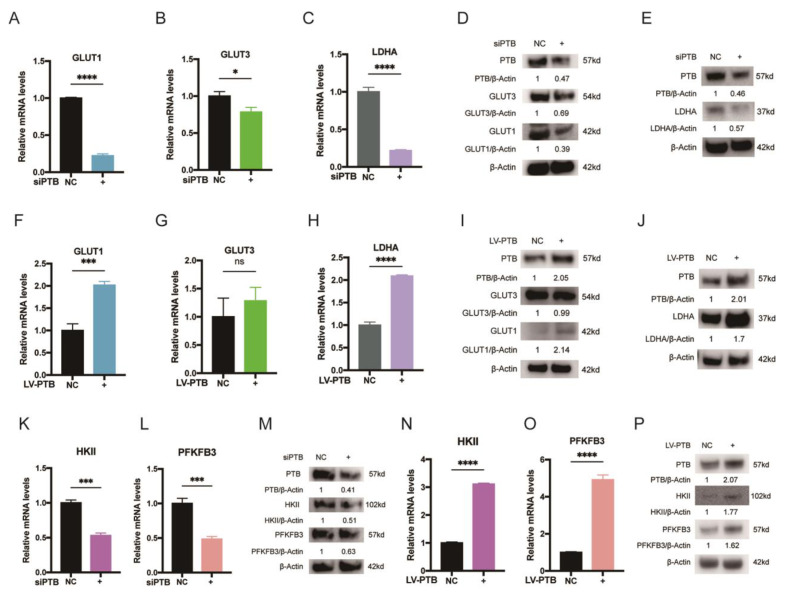
PTB promotes glycolysis enzymes in KFb. The mRNA (**A**–**C**) and protein (**D**,**E**) levels of GLUT1, GLUT3, and LDHA in the KFb transfected with siPTB were determined. The mRNA (**F**–**H**) and protein (**I**,**J**) levels of GLUT1, GLUT3, and LDHA in the NFb transfected with the PTB lentivirus were determined. The mRNA levels of HKII (**K**) and PFKFB3 (**L**) in KFb transfected with siPTB were investigated by qRT-PCR. (**M**) Their protein levels were determined by Western blotting. The mRNA level of HKII (**N**) and PFKFB3 (**O**) in NFb transfected with the PTB lentivirus were investigated by qRT-PCR. (**P**) Their protein levels were determined by Western blotting (*n* = 3). Data are presented as the mean ± S.D. * *p* < 0.05, *** *p* < 0.001, **** *p* < 0.0001.

**Figure 4 ijms-24-05162-f004:**
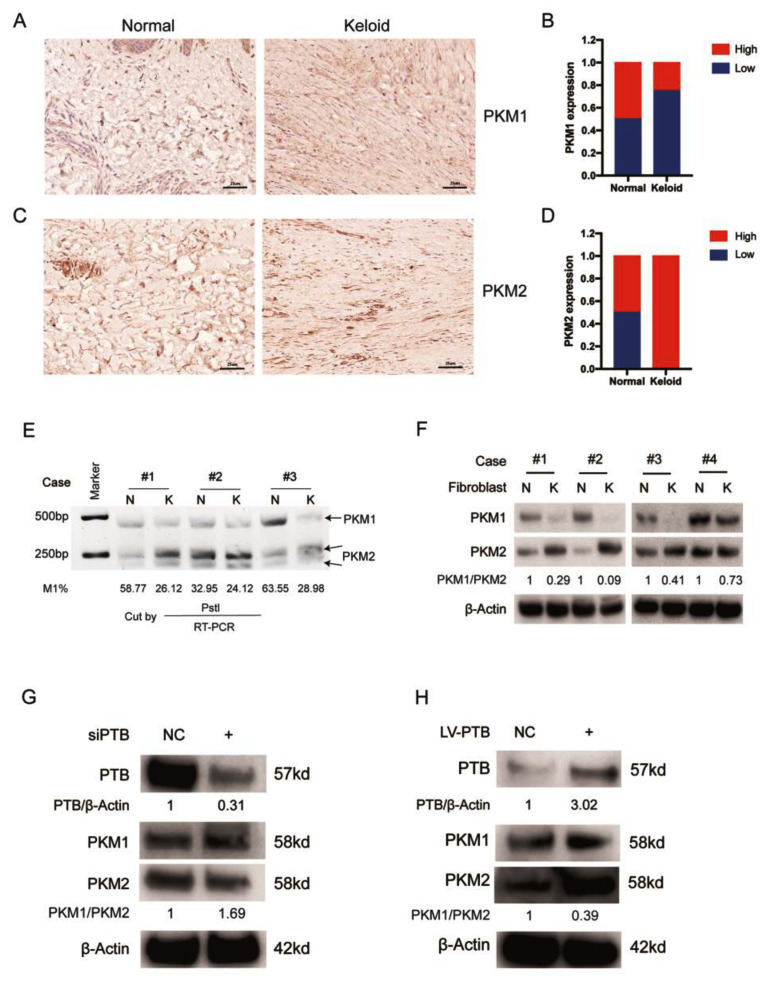
PTB regulates PKM2 splicing in KFb. (**A**) The expression of PKM1 was detected in keloid tissues and the corresponding normal skin tissues by IHC. (**B**) Differences in the expression scores of PKM1 between keloid tissues and the corresponding normal tissues are presented as a histogram (*n* = 4). (**C**) The expression of PKM2 was detected in keloid tissues and the adjacent normal skin tissues by IHC. (**D**) Differences in PKM2 expression scores between keloid tissues and the corresponding normal tissues are presented as a histogram (*n* = 4). (**E**) The PKM1 and PKM2 isoforms were detected using PstI and RT-PCR in the NFb (*n* = 3) and KFb (*n* = 3). (**F**) PKM1 and PKM2 protein levels in the KFb (*n* = 4) and the matched adjacent NFb (*n* = 4) were detected by Western blotting. (**G**) PTB silencing reduced the protein levels of PKM2 in KFb detected by Western blotting. (**H**) PTB overexpression in NFb increased the protein levels of PKM2. Data are presented as the mean ± S.D.

**Figure 5 ijms-24-05162-f005:**
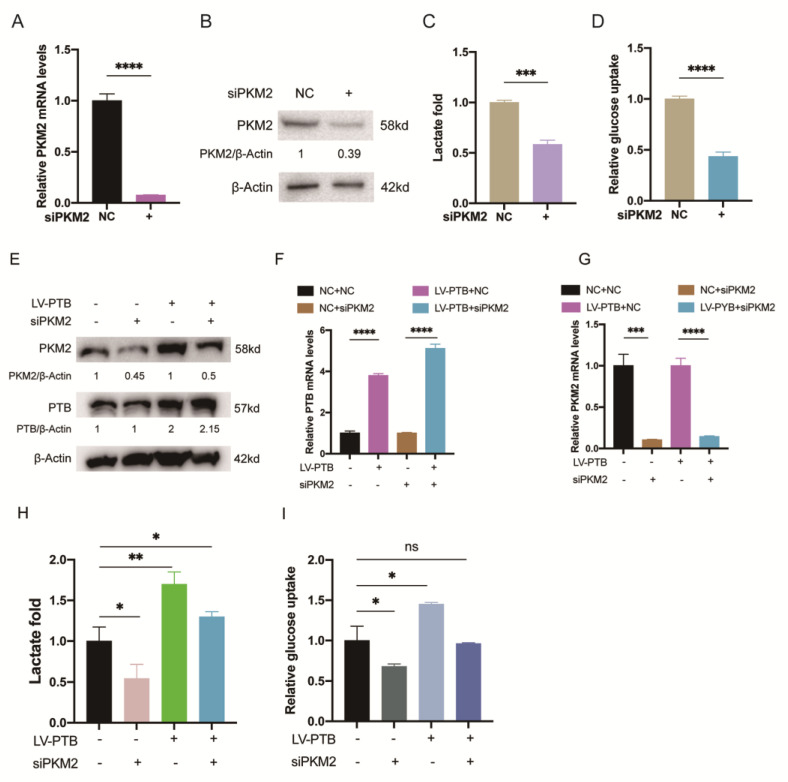
PTB regulates aerobic glycolysis through alternative splicing of PKM. (**A**,**B**) The efficiency of PKM2 silencing. Lactate production (**C**) and glucose uptake (**D**) were measured in the KFb transfected with siKM2. The overexpression efficiency of PTB and the silencing efficiency of PKM2 were confirmed by Western blotting (**E**) and qRT-PCR (**F**,**G**). Lactate production (**H**) and glucose uptake (**I**) were detected to measure the influence of knocking down PKM2 on the overexpression of PTB by NFb (*n* = 3). Data are presented as the mean ± S.D. * *p* < 0.05, ** *p* < 0.01, *** *p* < 0.001, **** *p* < 0.0001.

**Figure 6 ijms-24-05162-f006:**
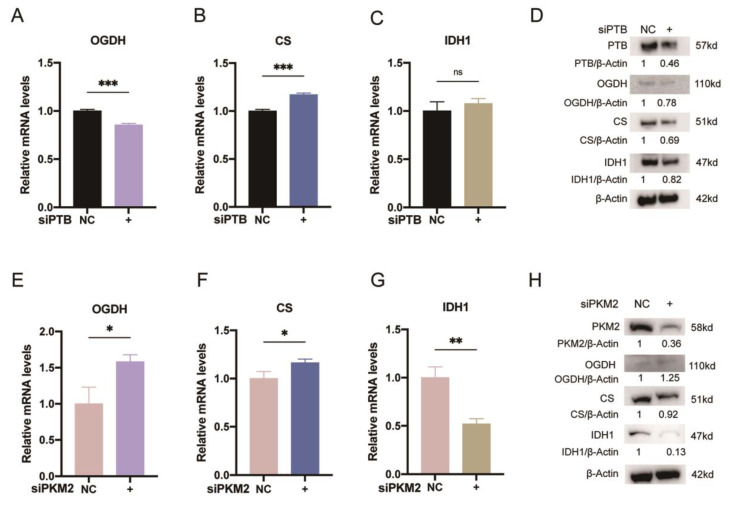
PTB knockdown downregulated the key enzymes of the TCA cycle. qRT-PCR (**A**–**C**) and Western blot (**D**) analyses of OGDH, CS, and IDH1 were performed after KFb were transfected with siPTB. qRT-PCR (**E**–**G**) and Western blot (**H**) analyses of OGDH, CS, and IDH1 were performed after KFb were transfected with siPKM2 (*n* = 3). Data are presented as the mean ± S.D. * *p* < 0.05, ** *p* < 0.01, *** *p* < 0.001.

**Figure 7 ijms-24-05162-f007:**
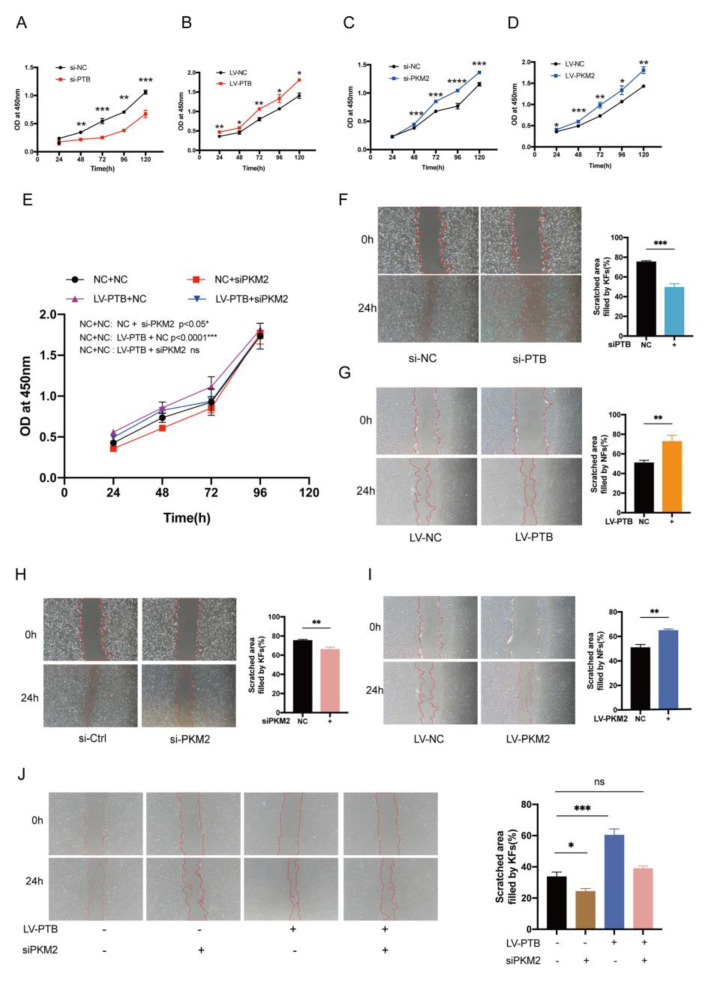
PKM2 knockdown abolished the PTB-induced proliferation and migration of fibroblasts. (**A**) CCK8 assay evaluation of the influence of the knockdown of PTB on the proliferation of KFb. (**B**) PTB overexpression enhanced the proliferation of NFb. (**C**) CCK8 assay of the influence of the knockdown of PKM2 on the proliferation of KFb. (**D**) Overexpression of PKM2 enhanced the proliferation of NFb. (**E**) CCK8 assay of the influence of the knockdown of PKM2 on NFb overexpressing PTB (*n* = 3). (**F**) Scratch-wound assay showing KFb transfected with si-NC migrating from the edges of scratches into vacant areas, with complete closure by Day 2, whereas KFb transfected with siPTB had slower cell growth and decreased migration, and vacant areas were not fully covered on Day 2. (**G**) Overexpression of PTB increased the migration capacity of NFb (*n* = 3). (**H**) CCK8 assay of the influence of knocking down PKM2 on the proliferation of KFb. (**I**) Effect of the overexpression of PKM2 on the proliferation of NFb. (**J**) PKM2 siRNA and PTB-overexpressing lentivirus were both transfected into NFb, and then the scratch wound assay was carried out (*n* = 3). Data are presented as the mean ± S.D. * *p* < 0.05, ** *p* < 0.01, *** *p* < 0.001, **** *p* < 0.0001.

**Table 1 ijms-24-05162-t001:** Sample information.

Sample	Site	Gender	Age (Years)	Period (Months)	Cause
Normal skin					
N1	Eyelid	Female	25		
N2	Eyelid	Female	27		
N3	Eyelid	Female	19		
N4	Eyelid	Female	29		
N5	Eyelid	Female	18		
N6	Eyelid	Female	23		
N7	Eyelid	Female	28		
N8	Abdomen	Female	36		
N9	Eyelid	Female	50		
N10	Hand	Female	35		
N11	Eyelid	Female	45		
N12	Eyelid	Female	19		
N13	Eyelid	Female	26		
N14	Eyelid	Female	33		
N15	Eyelid	Female	41		
N16	Eyelid	Female	46		
N17	Eyelid	Female	19		
N18	Eyelid	Female	27		
Keloid					
* K1	Perineum	Female	17	14	Scald
* K2	Shoulder	Female	21	23	Surgery
* K3	Chest	Male	26	32	Unknown
* K4	Ear	Male	24	22	Surgery
K5	Chest	Male	38	12	Surgery
K6	Neck	Female	44	19	Surgery
K7	Arm	Male	41	20	Surgery
K8	Chest	Female	52	13	Burn
K9	Shoulder	Female	46	16	Surgery
K10	Shoulder	Female	29	39	Unknown
K11	Chest	Male	25	13	Surgery
K12	Ear	Male	27	23	Unknown
K13	Shoulder	Male	39	30	Surgery
K14	Chest	Female	48	29	Surgery
K15	Shoulder	Female	29	19	Surgery
K16	Arm	Male	32	28	Burn
K17	Neck	Male	33	120	Surgery
K18	Shoulder	Male	45	32	Surgery
K19	Ear	Female	20	14	Ear piercing
K20	Ear	Female	39	12	Surgery
K21	Chest	Female	36	36	Surgery

The period represents the time (months) of keloid formation; * in the sample list represents four keloid samples that had adjacent normal skin tissue.

**Table 2 ijms-24-05162-t002:** Primer sequences.

Gene	Primer Sequence (5′ to 3′)
PTB F	ACGCACATTCCGTTGCCTTAC
PTB R	AACCTGCCTCTACAGCGTCCA
GLUT1 F	TGTGGGCArGTGCTTCCAGTA
GLUT1 R	CGGCCTTTAGTCTCAGGAACTTTG
GLUT3 F	GAGGTGCTGCTCACGTCTC
GLUT3 R	GAAACCGTCCGCGTTAAGTT
LDHA F	ATGGCAACTCTAAAGGATCAGC
LDHA R	CCAACCCCAACAACTGTAATCT
HKII F	GAGCCACCACTCACCCTACT
HKIIR	CCAGGCATTCGGCAATGTG
PFKFB3 F	TTGGCGTCCCCACAAAAGT
PFKFB3 R	AGTTGTAGGAGCTGTACTGCTT
PKM2 F	TGCCGTGGAGGCCTCCTTCAAGT
PKM2 R	GGGGCACGTGGGCGGTATCTG
OGDH F	GGCTACGTGTTGACGCCATA
OGDH R	CTCAACTTAGCAGCACAAGTCCTTA
CS F	GTCTGGCTAACACAGCTGCAGA
CS R	CATGGCCATAGCCTGGAACA
IDH1 F	AATCAGTGGCGGTTCTGTGGTA
IDH1 R	ACTTGGTCGTTGGTGGCATC
β-actin F	GTCCACCGCAAATGCTTCTA
β-actin R	TGCTGTCACCTTCACCGTTC

**Table 3 ijms-24-05162-t003:** Information on the antibodies.

Antibody	Dilution	Companies
PTB	1:1000 (WB)	Abcam, USA
PTB	1:100 (IHC)	Abcam, USA
PKM1	1:400	Novus, St. Charles, MO, USA
PKM2	1:1000	CST, Boston, MA, USA
GLUT1	1:1000	Abcam, USA
GLUT3	1:1000	Abcam, USA
LDHA	1:1000	CST, USA
PFKFB3	1:1000	Abcam, USA
HKII	1:1000	CST, USA
OGDH	1:1000	CST, USA
IDH1	1:1000	Abcam, USA
CS	1:1000	Abcam, USA
β-actin	1:1000	ZSGB-BIO, Beijing, China

## Data Availability

The data presented in this study are available upon request from the corresponding author.

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
