# Peer review of "PTB Regulates the Metabolic Pathways and Cell Function of Keloid Fibroblasts through Alternative Splicing of PKM"

_ijms, 2023, doi:10.3390/ijms24065162_

Round 1

Reviewer 1 Report

Introduction, line 61

Although mentioned in abstracts, please state the abbreviation when they first appear in the main text: PKM~PKL etc.

Results

1) Figure 1A

‘Normal’ seems to be ‘adjacent normal’, while 1B states ‘corresponding normal’ which can be confounding. Please state clearly whether both normal tissue states the same or not.  

Are all the normal samples are ‘adjacent normal skin’ throughout the article?

2) Figure 2

How many times were the experiment repeated?

Please state the repetition number and statistical significance on the Figure 2.

3) Figure 3

IHC on Figs 3A&C is not stated clearly.

Does PKM1 and PKM2 show cytosolic or nuclear stain?

Quantification of Fig3C presented in Fig3D does not matches the IHC stain positivity of Fig3C, which shows similar reactivity.

4) Figure 7

Image quality of migration assay (especially Fig7J) should be improved. Cell migration on the third and fourth column does not match the red line depicted by the authors.  

Discussion.

1)line 266 please correct the typo “Au Mounting”

Author Response

Dear reviewer,

Thank you so much for the careful and thoughtful revision advice, we really appreciate the valuable comments that you offered and made the following responses.

Result

1) Yes, both the adjacent normal tissue and the corresponding normal tissue stated in figure 1 state the same. And as we reported in the method part, there are four pairs of keloid and adjacent normal skin tissues of all the samples. The other 18 normal skin samples were obtained from patients who underwent cosmetic surgery.

2) The experiment was repeated three times. The repetition number was added in Figure 2.

3) When keloid tissues were compared to matched normal skin tissues, we found half of the normal tissues (50%, 1/2 cases) expressed high PKM1 level and the left samples expressed low PKM1 level. When it comes to the keloid tissues, 75% (3/4 cases) expressed low PKM1 levels. Unlike the PKM1 result, all of the keloid tissues expressed high levels of PKM2 in the keloid tissue while only half of the normal tissues expressed a high level of PKM2.

PKM1 and PKM2 show cytosolic stains.

PKM2 in Figure. 4C: Many of the stains in normal skin are non-specific, such as collagen fibers, for many of them show no nuclear stain in the picture. Thus, the positive cells in normal tissue are fewer than in the keloid tissue.

4)Figure 7. We are sorry for the unqualified images in figure7. However, we are experiencing serious covid-19 recently for the policy change. And the Spring festival is on the way, thus, we are afraid we can not shoot more high-quality images of them. Besides, we checked again our primary data and have not found a mismatch between the third and fourth columns in the histogram with the migration pictures. We would be grateful if you could give us more detailed instructions about this question.

Discussion.

1) The spelling mistake has been corrected in Mounting studies.

Reviewer 2 Report

Listed below are the point to consider in order to make the paper better.

1.     The authors need to show the histological analysis of keloid tissue for example H&E, Massion’s trichome, and Picrosirius red. They must show the character of the keloid tissue that they used in their study. Keloid fibroblast shows their unique character.

2.     The authors need to show the qPCR and immunofluorescence analysis of the keloid tissue that they used. These data need to confirm the keloid tissue.

3.     The authors need to explain the PTB expression of total keloid tissue and normal tissue. They told us 75% of keloid tissue demonstrated PTB overexpression and 25% of keloid tissue did not demonstrate PTB overexpression. Therefore, they need to explain that PTB overexpression is the unique characteristic of keloid tissue. If the PTB expression is decreased, the keloid tissue is normalized, and the character of keloid is disappeared.

4.     The authors need to explain that the normalization of keloid tissue is caused by the knockdown of PKM2. Their study is very well but they have to focus on not only the metabolism of keloid tissue but also the normalization of keloid character because the aim of keloid study improves the symptom of keloid scar in patients.

Author Response

Dear reviewer,

Thank you so much for the careful and thoughtful revision advice, we really appreciate the valuable comments that you offered and made the following responses.

  1. The character of the keloid tissue in HE: The dermis of keloid has different histological characters from superficial to deep layers. The collagen fiber bundles in the superficial layer are slightly smaller and more sparsely distributed than those in the middle and deep layers, and cell infiltration is much higher. The middle layer of the dermis is very dense, and the collagen fiber bundle is significantly thicker than the superficial layer. On the other hand, the collagen fiber bundles in the deep layer of the dermis clearly deteriorate, and there are relatively few cells there. The histological research is the background of our study. And considering this is a common experiment, thus, we did not include it in this manuscript.
  2. I feel so sorry that I didn’t understand the second suggestion because the qPCR and immunofluorescence of PTB results have been demonstrated in the results of Figure.2. We would be grateful if you can make this question clear to us.
  3. Actually, we only carried out an experiment on the PTB expression difference in keloid tissue and its corresponding normal skins, thus we could only come to the conclusion that PTB overexpression is the character of keloid. However, we think your suggestion is of great value, which encourages us to explore the PTB expression level in other types of scars.
  4. Yes, we also hold this opinion, thus we reported the result of figure 7, which is PKM2 knockdown abolished the PTB - induced proliferation and migration of fibroblasts. We think this also means the PKM2 knockdown resulted in the normalization of keloid.

Round 2

Reviewer 1 Report

The authors have well revised the manuscript. 

Author Response

Dear reviewer,

Thank you again for your kind work.

Reviewer 2 Report

1.     The keloid tissue shows the specific character in qPCR such as increasing collagen type 1, a-SMA, and Vimentin. The keloid tissue that they used needs to confirm using qPCR analysis.

Author Response

Dear reviewer,

Thank you so much for the careful and thoughtful revision advice, we really appreciate the valuable comments that you offered and made the following response.

1) The keloid tissue shows the specific character in qPCR such as increasing collagen type 1, a-SMA, and Vimentin. The keloid tissue that they used needs to confirm using qPCR analysis.

Response: In our past primary study, we also carried out the qPCR experiment to confirm the genes related to the extracellular matrix. The results were reported in the paper named: TGF-β1 Induces Polypyrimidine Tract-Binding Protein to Alter Fibroblasts Proliferation and Fibronectin Deposition in Keloid. DOI: 10.1038/srep38033. Thus, we did not include it in this study.
